# Antinociceptive Effect of the Combination of a Novel α4β2* Agonist with Donepezil in a Chronic Pain Model

**DOI:** 10.3390/biomedicines11123249

**Published:** 2023-12-08

**Authors:** Fernanda B. de M. Monte, Tadeu L. Montagnoli, Bruno E. Dematté, Fernanda Gubert, Vitória S. Ventura, Jaqueline S. da Silva, Margarete M. Trachez, Rosalia Mendez-Otero, Gisele Zapata-Sudo

**Affiliations:** 1Programa de Pós-Graduação em Farmacologia e Química Medicinal, Instituto de Ciências Biomédicas, Universidade Federal do Rio de Janeiro, Rio de Janeiro 21941-617, Brazil; fe_monte@yahoo.com.br (F.B.d.M.M.); tmontagnoli@gmail.com (T.L.M.); ssjck@hotmail.com (J.S.d.S.); 2Programa de Pós-Graduação em Cardiologia, Instituto do Coração Edson Saad, Universidade Federal do Rio de Janeiro, Rio de Janeiro 21941-617, Brazil; brunodematte@outlook.com; 3Instituto de Ciências Biomédicas, Universidade Federal do Rio de Janeiro, Rio de Janeiro 21941-617, Brazil; fernanda.gubert@icb.ufrj.br; 4Instituto do Coração Edson Saad, Faculdade de Medicina, Universidade Federal do Rio de Janeiro, Rio de Janeiro 21941-617, Brazil; vitoriaventura@gmail.com (V.S.V.); mmtrachez@gmail.com (M.M.T.); 5Instituto de Biofísica Carlos Chagas Filho, Universidade Federal do Rio de Janeiro, Rio de Janeiro 21941-617, Brazil; rmotero@biof.ufrj.br

**Keywords:** chronic pain, spinal nerve ligation, nicotinic receptor, anticholinesterase, neuroinflammation

## Abstract

Chronic pain presents a major challenge in contemporary medicine, given the limited effectiveness and numerous adverse effects linked to available treatments. Recognizing the potential of the cholinergic pathway as a therapeutic target, the present work evaluates the antinociceptive activity of a combination of Cris-104, a novel α4β2* receptor agonist, and donepezil, a central anticholinesterase agent. Isobolographic analysis revealed that equimolar combination was approximately 10 times more potent than theoretically calculated equipotent additive dose. Administration of Cris-104 and donepezil combination (3 μmol/kg) successfully reversed hyperalgesia and mechanical allodynia observed in rats subjected to spinal nerve ligation (SNL). The combination also modulated neuroinflammation by reducing astrocyte activation, evident in the decreased expression of glial fibrillary acidic protein (GFAP) in the spinal cord. The observed synergism in combining a nicotinic receptor agonist with an anticholinesterase agent underscores its potential for treating chronic pain. This alternative therapeutic distinct advantage, including dose reduction and high selectivity for the receptor, contribute to a more favorable profile with minimized adverse effects.

## 1. Introduction

Pain originates from the stimulation of afferent nerve fibers (nociceptors), whose impulse is conducted from the periphery to the spinal cord. In physiological conditions, the excitability threshold of nociceptors is high; however, when tissue injury occurs or the intensity and frequency of the nociceptor activation increases, action potentials are generated and propagated to the dorsal horn of the spinal cord, inducing pain. Neuropathic chronic pain is caused by an injury or disease of the somatosensory nervous system and has a prevalence of 6–8% in the general population, 20–25% in diabetics, 35% in HIV-infected patients and 66% in the elderly [1,2]. In addition to the high prevalence, it is associated with a reduced quality of life due to cognitive deterioration, depression and obesity [3,4] resulting in socio-economic problems related to reduced labor performance [5]. The main mechanisms proposed to induce chronic pain include: 1. sensitization of nociceptors; 2. abnormal excitability of afferent neurons; 3. nociceptive facilitation of the dorsal horn of the spinal cord; 4. dysfunction of inhibitory pathways; 5. sympathetic nervous system action; and, 6. central neuroplasticity. The restricted therapeutic arsenal for the treatment of chronic pain is not satisfactory, due to low efficacy and side effects, and thus, identifying new targets is urgent [6,7]. There are many mechanisms involved in the genesis of neuropathic pain, making treatment extremely difficult. Only ~50% of patients obtain pain relief from currently available drugs, including anticonvulsants, such as gabapentin and pregabalin; local anesthetics, such as lidocaine; and tricyclic antidepressants, such as amitriptyline [8]. The modulation of pain perception depends on the release of excitatory neurotransmitters such as glutamate and substance P that lead to the depolarization of postsynaptic membranes. After central processing, pain stimuli can activate descending pathways in a negative-feedback loop, whose main function is to minimize its transmission. Neurotransmitters involved in the descending inhibitory pathways include endorphins, enkephalins, serotonin, and epinephrine, which appear to modulate pain. Membrane hyperpolarization or blockade of Ca^2+^ influx at primary afferent neurons is an important mechanism of pain relief, as observed with μ-opioid, α_2_-adrenergic, M2 muscarinic and serotoninergic receptor activation [9]. Therefore, the involvement of several pathways in pain transmission, such as the adrenergic, cholinergic, serotonergic and opioid, is important because it provides alternative pharmacological targets for the treatment of neuropathic pain. 

Recently, the perceived importance of the cholinergic pathway has increased because of the antinociceptive action produced by anticholinesterase agents (neostigmine and donepezil) and nicotinic agonists (epibatidine and ABT-594) [10]. Mammalian brain structures are modulated by the cholinergic system as they control descending inhibitory pathways to modulate excitability in the spinal cord by suppression of nociceptive ascending pathways [11]. Donepezil (DPZ) is a reversible and selective inhibitor of acetylcholinesterase (AChE) which can be safely administered to patients with mild to moderate liver and kidney disease. Its side effects, caused mainly by its cholinomimetic action, are usually mild and transient. Its mechanism of action involves increased gamma-aminobutyric acid (GABA) release and activation of GABAergic signaling. Epibatidine is an alkaloid isolated from amphibian skin extracts and shows two hundred times greater analgesic potency than morphine in acute pain assays resulting from the activation of α4β2* nicotinic acetylcholine receptor (nAChR). Due to low incidence of tolerance, epibatidine has great potential for treatment of chronic pain but its therapeutic index is very limited, due to lack of selectivity among nAChR subtypes, interfering with neuromuscular junction [8]. Its use is responsible for adverse effects including cholinergic hyperactivity (nausea, dizziness and vomiting) and cardiovascular and central nervous system (CNS) toxicity, becoming inappropriate for long-term clinical use.

Over the years, numerous nAChR agonists have been tested and ABT-594 has emerged with antinociceptive effects, which can be attenuated by pretreatment with neuronal nAChR antagonists such as mecamylamine and chlorisondamine [7]. Its analgesic activity in acute and neuropathic pain models was equivalent in efficacy to morphine and its potency was 30 to 100 times greater. ABT-594 showed a better therapeutic index but it showed adverse effects with the analgesic doses used in phase II clinical trials [8]. Nevertheless, ABT-594, in addition to epibatidine, has led to a renewed interest in identifying the role of nAChRs in pain.

Nicotinic receptor α4β2* is widely distributed in the brain, including the major nucleus of the raphe and thalamus, locus coeruleus (LC) and periaqueductal gray matter in the midbrain, which play an important role in the descending pain modulatory pathway. The new agonist of the α4β2* receptor, Cris-104 (1-{2-[5-(4-fluorophenyl)-1*H*-pyrazol-4-yl]ethyl}piperidine), has shown important antinociceptive action in an animal model of chronic pain. Systemic administration of Cris-104 caused an increase in neuronal activity in the LC, increasing NE release, which was not observed when mecamylamine, a non-selective nAChR antagonist was administered [12]. Cris-104 has a permeability of 13.6 ± 7.1 cm/s at pH 7.4 in an artificial membrane and its log P is 3.07, indicating a high lipid solubility with good penetration into biological membranes, including the blood–brain barrier. It has good oral bioavailability with log D_7.4_ 1–3; plasma protein binding of 53%; and clearance rate of 1.8 mL/min/kg. The discovery of new neuronal ligands of nAChR α4β2* is of great interest in the medical field due to their potential use in the treatment of pain, whether neuropathic and/or inflammatory. 

DPZ is a therapeutic alternative for the treatment of chronic pain, since it has a good safety profile in clinical practice, good oral bioavailability and low incidence of cholinergic adverse effects, due to its high selectivity for acetylcholinesterase [11,13]. DPZ could emerge as an important therapeutic alternative for pain relief when used in combination with other substances, seeking synergistic action.

Considering this, a combination of Cris-104 with DPZ could be useful for the treatment of neuropathic pain because of their action in the cholinergic pathway. Therefore, this work investigated the potential synergism of these substances, which could reduce the doses of each and, consequently, reduce the adverse effects and toxicity. 

## 2. Materials and Methods

### 2.1. General

All experimental protocols were conducted in accordance with the Animal Care and Use Committee at Universidade Federal do Rio de Janeiro (protocols 103/19 and 041/19). Male Swiss mice (19–32 g) and male Wistar rats (200–240 g) were housed at 24 °C under a 12 h light/12 h dark cycle with free access to food and water. Cris-104 and DPZ were synthesized and supplied by Cristália Produtos Químicos e Farmacêuticos Ltda (Itapira, São Paulo, Brazil).

### 2.2. Central Antinociceptive Activity—Hot Plate Test

Nociception in animals can be estimated only by examining their reactions. The experimental model using a hot plate is a test for the evaluation of centrally acting analgesic substances. Supraspinal responses are observed after thermal stimulus (52 ± 0.5 °C) when animals are placed on a hot plate (Letica 7406, Barcelona, Spain). The latency of the appearance of the responses (licking the paws or jumping) is determined as being indicative of the beginning of the nociceptive response to the thermal stimulus, suggesting the presence of pain. The maximum time to keep animals on the hot plate is 35 s, to avoid injuries. Before the administration of the substances, the control latency was determined, and subsequently, animals were randomly treated orally (gavaged) with either Cris-104 or DPZ at different doses (3–100 µmol/kg). Latency for response was determined at 5 to 120 min after oral administration and data were expressed as maximum percentage of the effect (%MPE), which indicated the analgesic activity and was calculated using the following equation [14]:(1)$$\%MPE=\frac{\left(observed\;latency-control\;latency\right)\;(s)}{35-control\;latency\;(s)}\times 100\;$$

Dose–response curves were obtained using the results from the hot plate test (n = 10 mice/dose). Nonlinear regression of the dose–response curve provided the half maximal antinociceptive effect (ED_50_) for Cris-104 and DPZ. 

### 2.3. Isobolographic Analysis

Isobolographic analysis was used to determine the interaction between Cris-104 and DPZ, i.e., whether their effect could be consequence of synergism. In a cartesian coordinate system, the ED_50_ of Cris-104 was placed on the abscissa while the ED_50_ of DPZ was placed on the ordinate of the graph. The line connecting both ED_50_ is called the theoretical additive line. The dose required to cause half-maximal analgesic effect in the case of an additive interaction is called the additive theoretical point (ED_50,add_), which was determined using the following equation: *ρ*_1_ = molar fraction of Cris-104; *ρ*_2_ = molar fraction of DPZ; *R* = potency scale [11]
(2)$${ED}_{50,\;add}=\frac{{ED}_{50,\;Cris-104}}{\left(\rho_{1}+R\rho_{2}\right)}$$
(3)$$R=\frac{{ED}_{50,\;Cris-104}}{{ED}_{50,\;DPZ}}$$

Subsequently, mice were orally treated with the combination of Cris-104 and DPZ in a equimolar proportion and final dose of 0.6, 1.2, 2.4 and 4.8 μg/kg. The dose-response curve was constructed and ED_50_ for the combination was determined (ED_50,mix_). A case in which the ED_50,mix_ is below the ED_50,add_ indicates that the effect of the combination results from synergism. 

### 2.4. Chronic Pain Model—Spinal Nerve Ligation

A spinal nerve ligation (SNL) model was chosen to induce injury to produce chronic pain. After anesthesia with ketamine (80 mg/kg i.p.) and xylazine (15 mg/kg i.p.), male Wistar rats were prepared for the surgery and ligation of L5 spinal nerve was performed. The animals were randomly divided into groups: one which had L5 ligation, and another submitted to same procedure, except for nerve injury (Sham). After surgery and confirmation of neuropathic pain, rats were treated with the combination of Cris-104 + DPZ at a dose three times that of DE50mix, 3.2 μmol/kg. Thus, animals received orally (gavaged) the combination of Cris-104 (1.6 μmol/kg) + DPZ (1.6 μmol/kg) or saline. 

### 2.5. Behavioral Tests of Thermal Hyperalgesia and Mechanical Allodynia

Thermal hyperalgesia was evaluated using an analgesimeter (model 37370, Ugo Basile, Milan, Italy) through the determination of the latency of paw removal during the exposure to a radiant heat source on its surface. The paw-withdrawal reflex automatically interrupts the light beam and the heat irradiation, thus obtaining the time latency of paw removal. The mean of three measurements was defined as the control latency, and the maximum time that the animal was exposed to the heat source (cutoff) was stated as 3 times the control time in order to avoid tissue damage. Mechanical allodynia was evaluated using a digital analgesimeter (model EFF 301, Insight, Sao Paulo, Brazil) to quantify changes in tactile sensitivity in response to a mechanical stimulus. This electronic analgesimeter consisted of a pressure transducer adapted to a digital force counter expressed in grams. After 30 min of adaptation, pressure was applied to the center of paw until removal. The behavioral tests of thermal hyperalgesia and mechanical allodynia were performed before (baseline) and 7 days after experimental surgery for SNL and 3, 7, 10 and 14 days of treatment with combination of Cris-104 + DPZ.

### 2.6. Immunohistochemistry

At the end of the behavioral tests, animals under anesthesia with ketamine (80 mg/kg i.p.) and xylazine (15 mg/kg i.p.) were prepared for removal of the spinal cord, which was perfusion-fixed with 4% paraformaldehyde solution and submitted to cryopreservation by immersion in 10–30% sucrose solution gradient in PBS at 4 °C. Spinal cords were cut into 3 segments, L4, L5 and L6, embedded in Tissue-Tek O.C.T. Compound (Sakura Finetek, Torrance, CA, USA) and cut in cranial-caudal direction in 10 μm-sections using cryostat (model CM1850, Leica Biosystems, Nussloch, Germany). Spinal cord sections were incubated with primary antibody rabbit anti-GFAP (1:1000, ab7260, AbCam, Cambridge, MA, USA) overnight at 4 °C, and then for 60 min with anti-rabbit Alexa-488-conjugated secondary antibody (1:1000, Invitrogen, Waltham, MA, USA). After further washing with PBS, nuclei were stained with DAPI 0.5 μg/mL for 5 min, sections were mounted with VectaShield (Vector Labs, Newark, CA, USA) and analyzed under a confocal microscope (model LSM 510, Zeiss, Oberkochen, Germany) and quantified using ImageJ 1.54f software (NIH, Bethesda, MD, USA). 

### 2.7. Statistical Analysis

Analysis of variance (ANOVA) was used to compare multiple data with parametric distribution, followed by the Dunnett test (*post hoc*). The comparison between two experimental groups was performed using the paired or unpaired *t*-student test. Data were expressed as mean ± SEM and *p* < 0.05 were considered statistically significant.

## 3. Results

### 3.1. Antinociceptive Activity of the Combination Cris-104 + DPZ

Initially, the antinociceptive activity was investigated using the hot plate test after oral administration of DPZ (3, 5, 10, 30 μmol/kg) and Cris-104 (5, 10, 30 and 100 μmol/kg) alone. DPZ or Cris-104 showed antinociceptive activity in a dose-dependent manner. The MPEs were 10.3 ± 5.2; 22.3 ± 4.6; 38.6 ± 9.3 and 64.4 ± 8.5% after 15 min of DPZ treatment at doses of 3, 5, 10 and 30 μmol/kg, respectively (Figure 1A). 

A similar result was observed with the administration of Cris-104, which also showed antinociceptive activity with MPE of 7.0 ± 3.2; 29.1 ± 8.0; 38.7 ± 6.5; 37.9 ± 10.2% with the administration of 5, 10, 30 and 100 μmol/kg, respectively (Figure 1B). In a subsequent step, the central antinociceptive effect was also observed, using the equimolar combination of Cris-104 + DPZ at doses of 0.6; 1.2; 2.4 and 4.8 μmol/kg, with MPEs of 13.4 ± 7.1; 30.3 ± 6.7; 36.7 ± 5.3; 51.5 ± 7.4% after 15 min of treatment (Figure 1C). The MPE was greater (51.5 ± 7.4%) when animals were treated with the combination of Cris-104 + DPZ (4.8 μmol/kg), compared to 7.0 ± 3.2 and 29.1 ± 8.0% induced by DPZ or Cris-104 alone (5 μmol/kg). The calculated ED_50,mix_ was 1.07 ± 0.43 μmol/kg. 

Isobolographic analysis was performed to determine the type of interaction between Cris-104 and DPZ, which consisted of the construction of a graph which shows the ED50 of Cris-104 plotted on the *y*-axis and the ED_50_ of DPZ on the *x*-axis. Since the value of ED_50,mix_ is located below the additive line, it indicates that the combination Cris-104 + DPZ produces synergism (Figure 1D).

### 3.2. Combination Cris-104 + DPZ in Chronic Pain

Chronic pain was induced with SNL in rats that produced hyperalgesia and allodynia within 7 days after experimental surgery. Thermal hyperalgesia was characterized by a reduction in paw removal latency from 26.1 ± 1.6 to 14.7 ± 2.1 s (Figure 2A), while mechanical allodynia was detected by reducing the paw removal threshold from 49.2 ± 3.0 to 25.9 ± 2.4 g (Figure 2B). The animals that received only saline showed no improvement in these parameters, remaining altered until the end of the observation. However, the animals that presented thermal hyperalgesia induced by SNL, and that were treated with the combination of 3.4 μmol/kg of Cris-104 + DPZ, had total recovery after 2 weeks of treatment, recovering from 14.7 ± 2.1 to 22.5 ± 1.3 s (Figure 2A). Similarly, the mechanical allodynia observed after SNL was also reversed with the treatment of the combination of Cris-104 + DPZ (Figure 2B). After 2 weeks of treatment, the paw removal threshold recovered from 25.9 ± 2.4 to 42.2 ± 2.9 g (Figure 2B).

### 3.3. Influence of the Combination of Cris-104 + DPZ on the Glial Activation of Chronic Pain

In an attempt to verify the role of the activation of nicotinic receptors due to the association of Cris-104 + DPZ with the activation of glial cells and, consequently, the greater production of cytokines such as TNFα, the presence of glial fibrillary acid protein (GFAP) in the ipsilateral dorsal horns from the different experimental groups was investigated. An increased expression of GFAP was observed in rats submitted to SNL treated with the vehicle when compared to the Sham animals, which indicates greater astrocyte activation (Figure 3). In contrast, the treatment with the combination provided a reduction in GFAP expression in astrocytes, observed with the smaller marked area of this antibody in the spinal cord of animals submitted to treatment. As shown in Figure 3, the SNL led to an increase in the marked area, while the treatment normalized this expression.

## 4. Discussion

The use of the cholinergic pathway as a pharmacological target and a new strategy for the treatment of chronic pain has been gaining strength after the discovery of epibatidine because of its potential antinociceptive effect. Due to the involvement of nicotinic receptors in pain modulation, new ligands were designed and synthesized, including a pyazole analogue named Cris-104, which demonstrated specific binding to the nAChR (α_4_)_3_(β_2_)_2_ stoichiometry, exhibiting both a high affinity (61% of displacement) and low affinity to cytisine [15].

The antinociceptive activity of 34.4 ± 7.1% induced by Cris-104 (30 mg/kg) was comparable to the effect of morphine (10 mg/kg). This effect was attributed to the activation of the nicotinic receptor, confirmed by its reversal in the presence of selective α4β2* receptor antagonists, dihydrobetaerythroidin and mecamilamine [12]. Although promising, concerns emerged regarding potential adverse effects similar to those observed with ABT 594, another selective α4β2* agonist [16]. In an effort to mitigate the risk of adverse effects and reduce the dose used to promote the antinociceptive effect, the proposed strategy involved the combination of two agents, which could produce synergism in the activation of cholinergic pathway.

Our hypothesis was that there would be an improvement in pain relief when rats were administered the combination of Cris-104 with DPZ, a centrally acting anticholinesterase, utilized in clinical practice. DPZ is well tolerated after oral administration and has a well-established safety profile, with adverse gastrointestinal and cardiovascular effects similar to placebo, even in elderly patients [17]. The use of DPZ for pain management has been demonstrated in previous studies, in animal models, at doses of 1, 2 and 4 mg/kg (2.63, 5.27 and 10.54 μmol/kg) [18] or in association with gabapentin [19].

In the present study, the ED_50_ determined for DPZ was 7.8 ± 1.8 μmol/kg in a model of acute pain similar to that observed in a model of chronic pain induced by SNL [20]. The combination of Cris-104 and DPZ resulted in a reduction of 1/10 of the dose, with ED_50,mix_ equal to 1.07 μmol/kg. The synergism of Cris-104 and DPZ likely results from the fact that both act on the cholinergic pathway. DPZ could be responsible for increasing the availability of ACh in the synaptic cleft, due to the inhibition of AChE and enabling this neurotransmitter to activate muscarinic and nicotinic receptors in the brain and spinal cord, especially the α4β2* receptor, a target of Cris-104.

Synergism could be an interesting tool in the drug development process, reinforced by the fact that the dose used of each substance was highly reduced to achieve the antinociceptive effect in our acute pain model. However, it was necessary to investigate whether the same effect could be observed in a chronic pain model. Complete reversal of SNL-induced hyperalgesia and allodynia was detected with prolonged treatment of DPZ [21] or Cris-104 (30 mg/kg= 109 μmol/kg) [12]. Similarly, a total reversal of mechanical allodynia and thermal hyperalgesia was observed after treatment with the combination of 1.6 μmol/kg of DPZ and 1.6 μmol/kg of Cris-104. Synergism occurred in the chronic pain model, and it is important to identify the possible mechanisms involved. It can be highlighted that, the greater availability of ACh due to the lower degradation by AChE induced by DPZ could enable a greater activity of this neurotransmitter in the muscarinic receptors. DPZ could activate the M2, M3 and M4 receptors on spinal cholinergic neurons which, in turn, induce the release of the inhibitory neurotransmitter GABA [11]. This antinociceptive action can be explained not only by the GABAergic inhibitory action, but also by the action on presynaptic muscarinic receptors M2 and M4 in afferent primary neurons. The activation of M4 receptors is particularly interesting because this reduces the intracellular Ca^2+^ concentration and, consequently, lowers the release of glutamatergic excitatory neurotransmitters [22,23]. In addition, ACh can activate postsynaptic receptors, which reduces the excitability of the second-order neurons, promoting the activation of potassium channels (GIRK) and reducing the facilitation of the transmission of impulses [24]. The mechanisms by which nicotinic receptors interfere with nociceptive pathways are not yet fully elucidated. These receptors are found in suprasegmental and segmental regions, in ascending and descending pathways and therefore involved in pain modulation. In the spinal cord, the presence of nicotinic receptors α7 and α4β2* nAChR in cholinergic interneurons may contribute to analgesic and anti-inflammatory action [24]. Activation of α4β2* receptors by both ACh and Cris-104, which interferes with downward inhibitory modulation and is responsible for its antinociceptive activity [12]. Its action on α4β2* receptors at the LC, activating descending noradrenergic inhibitory pathways, promotes the local release of NE in the dorsal horn of the spinal cord. Combination of Cris-104 and DPZ could result in the release of NE, synaptic modulation in the dorsal horn of the spinal cord, activating cholinergic interneurons with GABAergic activation, and reduced excitation of primary glutamatergic afferent neurons. Additionally, Cris-104 has a noradrenergic action which is mediated by α_2_ receptors since antinociceptive activity is blocked in the presence of an α_2_-adrenergic antagonist such as yohimbine [12]. Similarly, the highest level of noradrenaline can be achieved by the inhibition of NE by Cris-104, similarly to gabapentin.

DPZ has a neuroprotective effect in chronic pain models through activating the MAPK pathway, promoting mitochondrial biogenesis and increasing cellular energy availability by increasing ATP production [18]. Although spinal and supraspinal mechanisms appear to be related to the process of pain modulation, it is considered that neuroinflammation is directly implicated in the process of persistent neuropathic pain. There is increased expression of glial cells such as microglia and astrocytes in the spinal cord, and the activation of astrocytes leads to the increased expression of GFAP and astrogliosis. These alterations occur due to the release of pro-inflammatory cytokines and chemokines in the posterior horn of the spinal cord, resulting from repeated noxious stimuli from peripheral nociceptors. The activation of astrocytes can be rapid (minutes), interfering in signaling via the phosphorylation pathways, intracellular calcium mobilization and translational regulation, or it can be long-lasting (hours or days), with changes in protein transcription and cell morphology, such as astrogliosis and hypertrophy. The accumulation of activated astrocytes in the spinal cord increases the release of pro-inflammatory cytokines (IL-1β and IL-6). As a result, there is an increased release of glutamate and increased binding to NMDA receptors, which contribute to postsynaptic neuronal depolarization and increased nociceptive signal. In addition, increased macrophages and activated microglia promote an increase in the expression of inflammatory markers, which are responsible for the continuous pain condition. Agonism of α4β2* receptor could reverse macrophage activation induced by neuronal insult through an independent mechanism of action that does not involve the descending noradrenergic pathways [25]. Activation of the α7 receptors in microglia also reduces TNFα expression [26]. DPZ could reduce inflammatory cells infiltration and microglia activation. When activated, astrocytes significantly enlarge the production of GFAP, an important marker of astrocyte activation [27]. Spinal cordsfrom animals submitted to SNL and treated with the combination of Cris-104 and DPZ, exhibited reduced expression of GFAP, indicating a reduction in the astrocyte activation process. The interference of the combination of Cris-104 and DPZ in neuroinflammation demonstrated in the SNL model can be compared to that observed with the use of DPZ alone in another animal model, in which it directly inhibited the inflammatory activation of microglia, reducing neuroinflammation and reducing the production of NO and TNFα. Despite the similarities in the effects observed in the different models, it should be considered that the dose of DPZ used alone was much higher than that used in the combination with Cris-104. That is a great advantage since high doses could limit the clinical use of DPZ in the treatment of chronic pain. Similar results were observed in previous clinical studies, which have shown that the association of DPZ with gabapentin had a superior analgesic effect than the use of gabapentin alone, and had similar side effects [28].

In the drug development process, it is essential to investigate potential adverse effects. Clinical use of DPZ is safe with low toxicity [17]. Nicotinic agonists previously tested have failed due to their narrow therapeutic range. Use of nicotinic receptor agonists can produce increased blood pressure, nausea and vomiting, which may occur due to the action of the ganglionar α4β2* receptor. No behavioral, motor activity or cardiovascular alterations were observed in animals treated with our combination. Administration of Cris-104 and DPZ exhibits synergistic effects in antinociception, enabling the use of a significantly reduced dosage and, consequently, minimizing the occurrence of side effects (Figure 4).

## 5. Conclusions

The combination of Cris-104 and DPZ produced synergism for antinociceptive action, providing a possible reduction in the dose required and a consequent reduction in side effects. DPZ could increase ACh availability though inhibition of AChE, and this neurotransmitter, along with Cris = 104, activates the muscarinic and nicotinic receptors, especially the α4β2* receptor. Thus, the combination prevented pain through the activation of inhibitory descending pathways; activation of the α4β2* receptor at the locus coeruleus; and a reduction in neuroinflammation, with less activation of microglia and astrocytes.

## Figures and Tables

**Figure 1 biomedicines-11-03249-f001:**
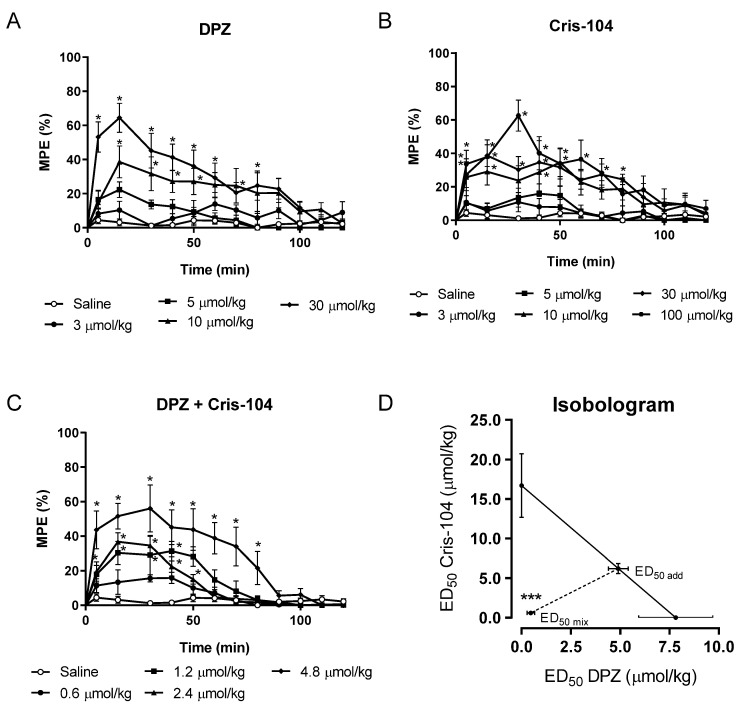
Antinociceptive effect of Cris-104 and donepezil (DPZ) p.o. on hot plate test. The %MPE induced by 3, 5, 10, and 30 µmol/kg of DPZ (**A**) or Cris-104 (**B**). (**C**) The %MPE induced by the co-administration of 0.6, 1.2, 2.4, and 4.8 µmol/kg of an equimolar mixture of DPZ and Cris-104. (**D**) Isobologram showing the interaction of Cris-104 with DPZ after oral co-administration in the rat. The additive line was drawn by connecting the value of ED_50_ for Cris-104 (ordinate) to the ED_50_ for DPZ (abscissa). ED_50,add_ = theoretical additive ED_50_; ED_50,mix_ = experimental ED_50_. Statistical analysis using Student’s *t*-test: * *p* < 0.05; *** *p* < 0.001.

**Figure 2 biomedicines-11-03249-f002:**
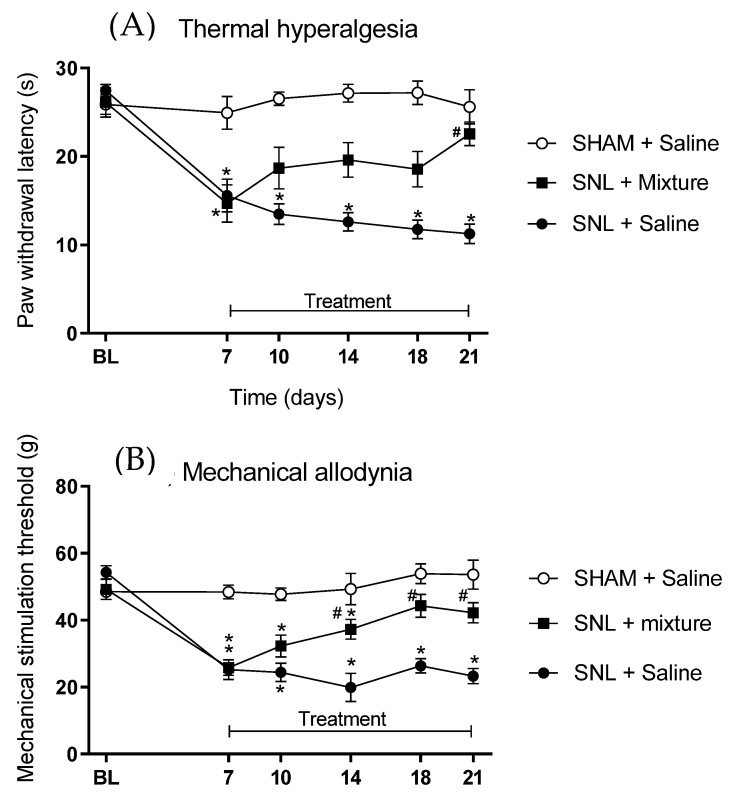
Improvement in spinal nerve ligation (SNL) induced hyperalgesia and allodynia after oral treatment with Cris-104 and DPZ. Thermal and mechanical paw withdrawal tests were performed before and after SNL surgery, and 3, 7, 10 and 14 days post treatment (n= 6 animals each group). (**A**) Time course of thermal paw withdrawal latency (s) before and after treatment. (**B**) Time course of mechanical stimulation threshold (g) before and after treatment. * *p* < 0.05 versus Sham + saline, # *p* < 0.05 versus SNL + saline; BL: baseline.

**Figure 3 biomedicines-11-03249-f003:**
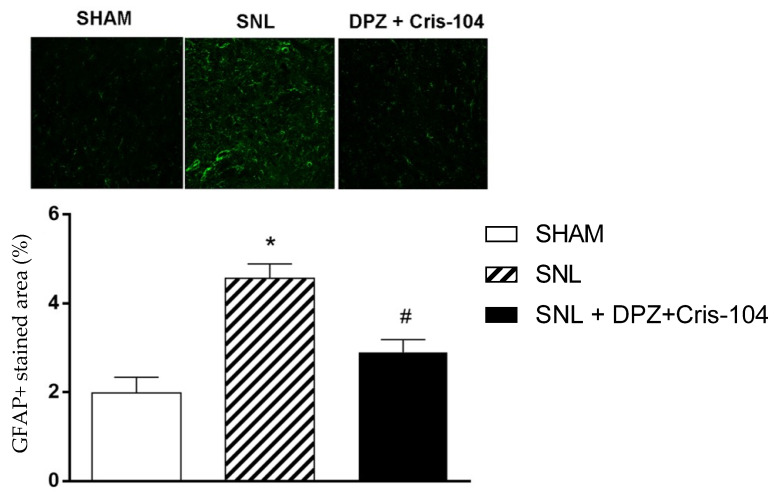
Immunofluorescence analysis of GFAP in spinal cords 14 days after spinal nerve ligation (SNL). GFAP expression was significantly increased in the SNL group compared with the sham and SNL + DPZ + Cris-104 groups. Administration of DPZ + Cris-104 attenuated SNL-enhanced GFAP expression. Data represent the mean ± SEM for 3 rats per group. * *p* < 0.005 compared to Sham; # *p* < 0.05 compared to SNL + saline. Scale bar, 50 μm; magnification, 400×.

**Figure 4 biomedicines-11-03249-f004:**
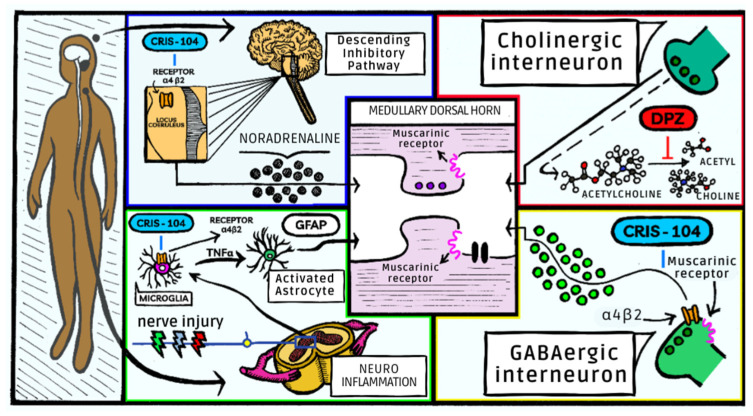
Signaling pathways involved in the effects of Cris-104 + DPZ combination.

## Data Availability

The data presented in this study are original and available on open access.

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
