# Peer review of "Antinociceptive Effect of the Combination of a Novel α4β2* Agonist with Donepezil in a Chronic Pain Model"

_biomedicines, 2023, doi:10.3390/biomedicines11123249_

Round 1

Reviewer 1 Report

Comments and Suggestions for Authors

The paper by Fernanda B. Monte et al., submitted to Biomedicines,  addresses an important topic that should be of interest for many readers in the field. Starting from relevant considerations about the urgent need for novel drugs to treat severe pain (including neuropathic chronic pain) the authors focus on the modulation of cholinergic pathways and evaluate the effects of two different agents, the central anticholinesterasic donepezil and a novel α4β2 receptor agonist. By different experimental approaches, they show that combination of the two compounds (both endowed with antinociceptive properties) is highly effective and exhibits synergism, and this could be a great advantage in view of possible therapeutic use. Moreover, interesting discussion and explanations are given. Although the work is interesting and scientifically sound ,  Reviewer found several important points that need to be addressed or corrected before publication of the study.

 Major comments :

1.       Abstract : line 20,   “….. 11 times lower than the theoretical dose calculated”. It is not clear to me how this precise value was calculated . Perhaps this part of the sentence can be substituted with “…..about 10 times lower than the theoretical dose calculated” , or please briefly explain how just the value “ 11” was obtained.

  2.       Page 3, line 130 . The whole Section 2.7 “Western blot” is referred to WB experiments that are not present in the result Section and therefore are not in this study. Please correct, or add the missing results (and discussion) if any.  Accordingly, if Section 2.7 will be deleted, in the ABSTRACT  section the related sentence (lines 22-24) must be corrected since the expression of TNF was not determined,  and also the involvement of microglia is not experimentally assessed in this study.

3.       Page 4, line 145. “Analysis of variance (ANOVA) was used to compare multiple data with nonparametric distribution, followed by the Dunnett test (post hoc).”  This sentence is not clear from my point of view.  To the best of my knowledge, ANOVA is a “parametric” analysis, and the post-hoc Dunnett’s test is a parametric test; generally it would be better if data with nonparametric distribution are analysed by nonparametric tests.  I think this point should be clarified or better explained. 

4.       Line 161 : “both the intensity and duration of the effect were increased with the administration of the combination”…. : by observation of  Figures 1c , Figures 1a and 1b  the statements of lines 161 to 164 seem rather difficult to be understood from the reader. Especially it is not clear how it is possible to clearly understand that intensity and duration are increased and I think that improved explanation is necessary.

5.       Line 188-189 : although qualitatively the meaning of the data reported is clear, it is not clear whether the numerical data really reflect the data reported in Figure 2A. For instance, a recovery up to 22.5  seems lower than in the Figure 2A.  Moreover I think “22.5 ± 1.3 g” is wrong because in Figure 2A “s” instead of “g”  is measured ?

6.       Line 192: again the numerical data (42.2 ± 2.9 g ) appear different from those in Figure 2B ! Please explain or correct

7.       Figure 2.  The time courses in Figure 2A might be in part wrong : the plot related to the “combination of drugs” (black squares) exhibits a very quick recovery from hyperalgesia starting from “day 7” to “day 10”, while in the text it is stated that the recovery was complete after 2 weeks of treatment (line 188), not after three days only. This is to be clarified, and the plot has to be checked carefully. 

8.       Figure 2: as stated also in Materials and Methods Section (lines 114-116) the treatment lasts two weeks. In the Figure it seems that treatment starts at day 10 and ends at day 21 … can the authors ameliorate presentation (or explain) ?

9.       Figure 2A : the symbols showing statistical significance (#) are close to the “sham” points (black circles) . In the Figure 2B instead the same symbols are close to the “treatment” symbols (black squares). I guess the Figure 2B is correct.. please correct or explain.

10.    Legend to the Figure 2 : Line 198 :  “p < 0.05 versus treated SNL” : please explain better which is the control ; from my point of view, “treated SNL” has no meaning here, or is very difficult. Please explain and/or correct. 

11.    Legend to Figure 3, Line 215 :  “ p < 0.05 compared with the LNE + DPZ + Cris-104 group”.  Please check this statement.

 12.    From line 220 (Discussion Section): many references are missing; at least most of them are required. This applies to sentences of lines 220-224 , 224-227,  227-229 , 236-238, 238-240 , 267-268 (if possible), 282-284, 285-286, 287-289, 292-294, 294-295, 305-307.

 13.    Lines 252-254 : is the reversal of mechanical allodynia and thermal hyperalgesia  really “total”  ?  According to Figure 2B, in the case of allodynia this reversal is only partial.  Please explain or correct the sentences.

NOTE : End of the manuscript ,  the “Data Availability Statement” (usually required in the journal Biomedicines ) is missing.

           Suggestion : Lines 257 to 284 :  the mechanisms discussed in this part of the manuscript are very interesting, although in some way they may appear a little difficult for some readers. I think that providing an additional Figure that shows (schematically !) the main mechanisms described, if feasible, would be of great help to increase the value of the manuscript and to obtain a more “friendly”, likely more attractive, manuscript. I suggest it,  even though not mandatory.

Minor points:

1.       Introduction, line 44 : the reference “Kimura et al., 2013” does not appear in the final references list; moreover it should be provided as a number  . Please correct

2.       Introduction, line 47 : reference “Sudo et al, 2018” , has to be adjusted similarly (see 1)

3.       Line 48 , typo :  “locus coeruleus”.

4.       Line 56  typo :  “protocols”

5.       Lines 162-163 : please delete the word “that” or alternatively “which”

6.       Line 190, typo  :  the treatment “with” the combination

7.       Figures 2A  and 2B,  tick labels:  the meaning of  “BL”  should be stated in the legend of Figure 2, because it might be not immediately understood by the reader.

8.       Legend to Figure 3, Line 212 , please explain the meaning of “LNE”

9.       Lines 254-255 ,  sentence : “ These doses are equivalent 16 times lower than ……”  : it is not clear the source of data for this comparison , it is likely that citing the appropriate references here again would be very helpful for readers.

10.      Line 312 : I suggest to change the sentence into something like ”In drug development, investigation of adverse effects is essential” or similar.  

Comments on the Quality of English Language

A moderate editing of English language is  recommended

Author Response

Major comments:

  1. Abstract: line 20,   “….. 11 times lower than the theoretical dose calculated”. It is not clear to me how this precise value was calculated. Perhaps this part of the sentence can be substituted with “…..about 10 times lower than the theoretical dose calculated” , or please briefly explain how just the value “ 11” was obtained.

As suggested by the reviewer, the phrase has been altered to “Isobologram analysis revealed that the effective dose of the combination was approximately 10 times lower than the theoretically calculated dose”

  1. Page 3, line 130. The whole Section 2.7 “Western blot” is referred to WB experiments that are not present in the result Section and therefore are not in this study. Please correct, or add the missing results (and discussion) if any.  Accordingly, if Section 2.7 will be deleted, in the ABSTRACT  section the related sentence (lines 22-24) must be corrected since the expression of TNF was not determined,  and also the involvement of microglia is not experimentally assessed in this study.

The reviewer well noted the error in the manuscript regarding WB experiments. Section 2.7 deleted and abstract corrected.

  1. Page 4, line 145. “Analysis of variance (ANOVA) was used to compare multiple data with nonparametricdistribution, followed by the Dunnett test (post hoc).”  This sentence is not clear from my point of view.  To the best of my knowledge, ANOVA is a “parametric” analysis, and the post-hoc Dunnett’s test is a parametric test; generally it would be better if data with nonparametric distribution are analysed by nonparametric tests.  I think this point should be clarified or better explained.

The reviewer commented was considered and the phrase corrected.

Page 4, line 1: “Analysis of variance (ANOVA) was used to compare multiple data with parametric distribution, followed by the Dunnett test (post hoc)”.

  1. Line 161: “both the intensity and duration of the effect were increased with the administration of the combination”…. : by observation of  Figures 1c , Figures 1a and 1b  the statements of lines 161 to 164 seem rather difficult to be understood from the reader. Especially it is not clear how it is possible to clearly understand that intensity and duration are increased and I think that improved explanation is necessary.

The text has been altered.

Page 5, line 6:

MPE was greater (51.5 ± 7.4%) when animals were treated with the combination of Cris-104 + DPZ (4.8 μmol/kg), compared to 7.0 ± 3.2 and 29.1 ± 8.0% induced by DPZ or Cris-104 alone (5 μmol/kg).).”

  1. Line 188-189 : although qualitatively the meaning of the data reported is clear, it is not clear whether the numerical data really reflect the data reported in Figure 2A. For instance, a recovery up to 22.5  seems lower than in the Figure 2A.  Moreover I think “22.5 ± 1.3 g” is wrong because in Figure 2A “s” instead of “g” is measured ?

Authors apologize for the incorrect figure included in the manuscript. The updated Figure 2 has been added which is in accordance to numerical data reported.

  1. Line 192: again the numerical data (42.2 ± 2.9 g) appear different from those in Figure 2B ! Please explain or correct

Since the correct figure 2 has been added to manuscript, the data are now in accordance to the text.

  1. Figure 2.  The time courses in Figure 2A might be in part wrong : the plot related to the “combination of drugs” (black squares) exhibits a very quick recovery from hyperalgesia starting from “day 7” to “day 10”, while in the text it is stated that the recovery was complete after 2 weeks of treatment (line 188), not after three days only. This is to be clarified, and the plot has to be checked carefully.

Since the altered figure 2 has been added to manuscript, the data are now in accordance to the text.

  1. Figure 2: as stated also in Materials and Methods Section (lines 114-116) the treatment lasts two weeks. In the Figure it seems that treatment starts at day 10 and ends at day 21 … can the authors ameliorate presentation (or explain) ?

Duration of treatment stated in methods lasts 2 wks and altered figure 2 shows that.

  1. Figure 2A : the symbols showing statistical significance (#) are close to the “sham” points (black circles) . In the Figure 2B instead the same symbols are close to the “treatment” symbols (black squares). I guess the Figure 2B is correct.. please correct or explain.

Altered figure 2 better demonstrated the statistical analysis.

  1. Legend to the Figure 2 : Line 198 :  “p < 0.05 versus treated SNL” : please explain better which is the control ; from my point of view, “treated SNL” has no meaning here, or is very difficult. Please explain and/or correct. 

To clarify the understanding of figure 2, legend has been altered: “*p < 0.05 versus Sham + saline, #p < 0.05 versus LNE + saline”.

  1. Legend to Figure 3, Line 215 :  “ p < 0.05 compared with the LNE + DPZ + Cris-104 group”.  Please check this statement.

To clarify the understanding of figure 3, legend has been altered: “* p < 0.005 compared to sham; # p < 0.05 compared to LNE + saline.

  1. From line 220 (Discussion Section): many references are missing; at least most of them are required. This applies to sentences of lines 220-224 , 224-227,  227-229 , 236-238, 238-240 , 267-268 (if possible), 282-284, 285-286, 287-289, 292-294, 294-295, 305-307.

References inserted

  1. Lines 252-254 : is the reversal of mechanical allodynia and thermal hyperalgesia  really “total”  ?  According to Figure 2B, in the case of allodynia this reversal is only partial.  Please explain or correct the sentences.

Updated graph shows the statement of total reversal

NOTE: End of the manuscript, the “Data Availability Statement” (usually required in the journal Biomedicines) is missing.

           Suggestion : Lines 257 to 284 :  the mechanisms discussed in this part of the manuscript are very interesting, although in some way they may appear a little difficult for some readers. I think that providing an additional Figure that shows (schematically!) the main mechanisms described, if feasible, would be of great help to increase the value of the manuscript and to obtain a more “friendly”, likely more attractive, manuscript. I suggest it, even though not mandatory.

For better understanding, it was inserted the figure below at the conclusion

Minor points:

Introduction, line 44 : the reference “Kimura et al., 2013” does not appear in the final references list; moreover it should be provided as a number  . Please correct

Corrected

Introduction, line 47 : reference “Sudo et al, 2018” , has to be adjusted similarly (see

Corrected

Line 48 , typo :  “locus coeruleus”.

Corrected

Line 56  typo :  “protocols”

Corrected

Lines 162-163 : please delete the word “that” or alternatively “which”

Deleted

Line 190, typo  :  the treatment “with” the combination

Corrected

Figures 2A and 2B, tick labels:  the meaning of  “BL”  should be stated in the legend of Figure 2, because it might be not immediately understood by the reader.

Altered

Legend to Figure 3, Line 212, please explain the meaning of “LNE”

Corrected

Lines 254-255 ,  sentence : “ These doses are equivalent 16 times lower than ……”  : it is not clear the source of data for this comparison , it is likely that citing the appropriate references here again would be very helpful for readers.

The phrase was deleted and text modified.

Line 312 : I suggest to change the sentence into something like ”In drug development, investigation of adverse effects is essential” or similar.

The phrase was altered:

In drug development process, it is essential to investigate potential adverse effects….

Comments on the Quality of English Language. A moderate editing of English language is recommended:

Manuscript checked for language improvement

Reviewer 2 Report

Comments and Suggestions for Authors

The importance of medications for treatment of chronic pain could hardly be overestimated, especially of non-opioid substances with high activity and lacking potential for development of tolerance and addiction. The example of potent antinociceptive cholinergic agonist epibatidine with exceptionally small safety margin prompted search for its analogs, retaining the antinociceptive but lacking the toxic properties of the frog alkaloid.

The described in the manuscript strategy to combine newly synthesized cholinergic agonist Cris-104 with an anticholinestherase agent Donepezil in order to decrease their dosages is interesting and the results are quite encouraging. However, the manuscript preparation is not up to the standards warranting its publication without substantial revisions.

First, the introduction to state of the art is quite insufficient. The story of epibatidine and search for its less toxic analogs (including Cris-104) is not presented (it is described in some more details in the first paragraph of the Discussion, but its place is in the Introduction).

Second, the methods are not well described. The details are fragmented/repeated in the subsections (e.g. in subsection 2.1 the operation of the rats is mentioned, but nothing is said what kind of operation it was before subsection 2.4; in the same subsection 2.1 the treatment  of the mice is not adequately described - it's not clear does they received Cris104+DPZ treatment).

Third, there are pitfalls in the results presentation. Administration of DPZ is described as 3, 10, 30 μmol/kg, and then the resulting MPE is described for DPZ treatment at doses of 5, 10 and 30 μmol/kg. The symbols on fig. 1 one are too small to compare them with the legend. The numbers provided for thermal hyperalgesia and mechanical allodynia on lines 189 and 192 does not look alike those plotted on fig.2.

Fourth, the discussion about the mechanisms of synergistic effects of Cris-104 and DPZ seems insufficient. The effect of this magnitude hardly could be explained by indirect, cross-transmitter interactions. Are there any data about metabolism of Cris-104, especially by AChE? If not, at least a speculation could be drawn about possibility of this type of interaction.

Fifth, in the Conclusions, the statement about "...resulting in the release of noradrenaline; synaptic modulation in the dorsal horn of the medulla, activating cholinergic interneurons with GABAergic activation and reduced excitation of primary glutamatergic afferent neurons..." does not follow from the presented experiments, and is appropriate in the Discussion not in Conclusions section.

Finally, the references presentation is very poor. The  Discussion almost lacks literature citations despite the presentation of a lot of literature data. The citation style is not uniform - in the Introduction, on lines 34, 36, 39, 44, and 47, four different citation styles are used. Some citations in the text are not included in the reference list (line 66 in the Materials and Methods).

Comments on the Quality of English Language

The manuscript needs extensive language editing. Some examples (a non-exhaustive list) follow:

 l44 - α4β2* - why asterisk?

l48 - locus cerullus - locus coeruleus?

l52 - potencial - potential?

l56 - experimetal ptotocols - experimental protocols?

l68 - when animals are placed in a hot plate - on a hot plate? many other occurences...

l71-72 - The maximum time of permanence in the hot plate - permanence? on the hot plate?

l82-85 - ...each ED50 of Cris-104 and DPZ were placed on the x and y axes, and the connected line, called theoretical additive line. - not clearly formulated

l92 - ...the effect of the combination results of synergism - results from synergism?

l123 - included in paraffin - embedded in paraffin?

l134 - using a spectrophotometer in the length of 750 nm - at a wavelength of 750 nm?

l235 - ...it [DPZ] has good tolerance to oral treatment... - DPZ is well tolerated after oral administration?

Author Response

First, the introduction to state of the art is quite insufficient. The story of epibatidine and search for its less toxic analogs (including Cris-104) is not presented (it is described in some more details in the first paragraph of the Discussion, but its place is in the Introduction).

The following paragraph has been included in Introdcution section:

“Epibatidine is an alkaloid isolated from amphibian skin extracts and shows analgesic potency two hundred times greater than morphine in the acute pain assays. Due to the low induction of tolerance, epibatidine had great potential for the treatment of chronic pain. However, its therapeutic index is very limited, due to the lack of selectivity in relation to the multiple subtypes of nicotinic acetylcholine receptors (nAChR) [8]. Despite the important analgesic activity resulting from the activation of nAChR, α4β2, this activity was not selective, interfering with the neuromuscular junction. Its use is responsible for adverse effects including cholinergic hyperactivity (nausea, dizziness, and vomiting) and cardiovascular and central nervous system (CNS) toxicity, becoming inappropriate for long-term clinical use. Over the years, numerous nAChR agonists have been tested and ABT 594 has emerged with antinociceptive effects, attenuated by pretreatment with neuronal nAChR antagonists such as mecamylamine and chlorisondamine [7]. Its analgesic activity in acute and neuropathic pain models was equivalent in efficacy to morphine and its potency was 30 to 100 times greater. ABT-594 showed a better therapeutic index, with slight side effects but although promising, it showed adverse effects with the analgesic doses used in phase II clinical trials [8]. Nevertheless, ABT-594, in addition to epibatidine, renewed interest in identifying the role of nAChRs in pain.”

Second, the methods are not well described. The details are fragmented/repeated in the subsections (e.g. in subsection 2.1 the operation of the rats is mentioned, but nothing is said what kind of operation it was before subsection 2.4; in the same subsection 2.1 the treatment  of the mice is not adequately described - it's not clear does they received Cris104+DPZ treatment).

The methods section has been reformatted:

Before the administration of the substances, the control latency was determined and subsequently, animals were randomly treated orally (gavage) with either Cris-104 or DPZ at different doses (3 – 100 µmol/kg). Latency for response was determined at 5 to 120 min after oral administration and data were expressed as maximum percentage of the effect (%MPE), which indicated the analgesic activity and was calculated using the following equation:

[12]

(1)

Dose-response curves were obtained using the results from the hot plate test (10 mice/dose). The linear regression of the dose-response curve provided the 50% of the maximum antinociceptive effect (ED50) for Cris-104 and DPZ.

Third, there are pitfalls in the results presentation. Administration of DPZ is described as 3, 10, 30 μmol/kg, and then the resulting MPE is described for DPZ treatment at doses of 5, 10 and 30 μmol/kg. The symbols on fig. 1 one are too small to compare them with the legend. The numbers provided for thermal hyperalgesia and mechanical allodynia on lines 189 and 192 does not look alike those plotted on fig.2.

Since the altered figure 2 has been added to manuscript, the data are now in accordance to the text.

Fourth, the discussion about the mechanisms of synergistic effects of Cris-104 and DPZ seems insufficient. The effect of this magnitude hardly could be explained by indirect, cross-transmitter interactions. Are there any data about metabolism of Cris-104, especially by AChE? If not, at least a speculation could be drawn about possibility of this type of interaction.

There is no information regarding the metabolism of this novel compound. After the demonstration of its pharmacological profile, the following step in the drug development process would be pharmacokinects and toxicity. Authors do not have enough information even for the description of a speculation.    

Fifth, in the Conclusions, the statement about "...resulting in the release of noradrenaline; synaptic modulation in the dorsal horn of the medulla, activating cholinergic interneurons with GABAergic activation and reduced excitation of primary glutamatergic afferent neurons..." does not follow from the presented experiments, and is appropriate in the Discussion not in Conclusions section.

This phrase added to discussion and deleted from conclusion

Finally, the references presentation is very poor. The  Discussion almost lacks literature citations despite the presentation of a lot of literature data. The citation style is not uniform - in the Introduction, on lines 34, 36, 39, 44, and 47, four different citation styles are used. Some citations in the text are not included in the reference list (line 66 in the Materials and Methods).

References reformatted and the missing citations included in the list

Comments on the Quality of English Language

The manuscript needs extensive language editing.

Manuscript checked for language improvement

Some examples (a non-exhaustive list) follow:

 l44 - α4β2* - why asterisk?

The asterisk is the form to describe the alpha4beta2* receptor which exhibits insensitivity to a-bungarotoxin and is the most widely distributed nicotinic receptor (nAChR) in the brain. Comprising five subunits, it can adopt two distinct conformations: (a4)2(B2)3 and (a4)3(ß2)2, each with different affinities for acetylcholine. The asterisk (*) in its name represents both arrangements, independent of stoichiometry, forming functional pentameric channels. This duality in subtypes may introduce terminology ambiguity, emphasizing the importance of understanding their independent roles in encoding channels permeable to cations through a central pore.

l48 - locus cerullus - locus coeruleus? Corrected

l52 - potencial - potential? Corrected

l56 - experimetal ptotocols - experimental protocols? Corrected

l68 - when animals are placed in a hot plate - on a hot plate? many other occurences...Corrected

l71-72 - The maximum time of permanence in the hot plate - permanence? on the hot plate? Corrected

l82-85 - ...each ED50 of Cris-104 and DPZ were placed on the x and y axes, and the connected line, called theoretical additive line. - not clearly formulated

Phrase altered to ….”the ED50 of Cris-104 was placed on the abscissa while the ED50 of DPZ was placed on the ordinate of the graph”.

l92 - ...the effect of the combination results of synergism - results from synergism? Corrected

l123 - included in paraffin - embedded in paraffin? Corrected

l134 - using a spectrophotometer in the length of 750 nm - at a wavelength of 750 nm? Section deleted

l235 - ...it [DPZ] has good tolerance to oral treatment... - DPZ is well tolerated after oral administration? Corrected

Round 2

Reviewer 1 Report

Comments and Suggestions for Authors

All my comments and suggestions have been properly addressed. The manuscript is now much improved also with the help of the added Figure 4. I still have  one comment :

Page 5, lines 206-208 : The presentation of data is more clear now. However, it seems that the numerical value of 7.0 ± 3.2 is related to Cris -104 (Fig. 1B)  and not to DPZ,  while the value for DPZ shown in Figure 2A seems to be about 20 rather than 29.1 ± 8. Please clarify or correct if necessary.

Reminder

Data availability statement is still missing.

Author Response

The reviewer commented : "Page 5, lines 206-208 : The presentation of data is more clear now. However, it seems that the numerical value of 7.0 ± 3.2 is related to Cris -104 (Fig. 1B)  and not to DPZ,  while the value for DPZ shown in Figure 2A seems to be about 20 rather than 29.1 ± 8. Please clarify or correct if necessary."

The reviewer is correct that value is related to Cris-104. Values  of 7.0 and 29.1 were obtained after 15 min of administration of Cris-104 at doses of 5 and 10 umol/kg.  This is stated in the following phrase of the revised version: 

"Similar result was observed with the administration of Cris-104, which also showed antinociceptive activity with MPE of 7.0 ± 3.2; 29.1 ± 8.0; 38.7 ± 6.5; 37.9 ± 10.2% with the administration of 5, 10, 30 and 100 μmol/kg, respectively (Figure 1B)."

 Figure 2A shows data from the combination of Cris-104 and DPZ.

Thus, there is no mistake regarding the results.

Reviewer 2 Report

Comments and Suggestions for Authors

It seems that authors did a great job improving the manuscript, all my concerns were addressed well enough to justify its publication in "Biomedicines"

Author Response

Authors would like to thank the reviewer for the comments that improved the manuscript.